# Relating Climate, Drought and Radial Growth in Broadleaf Mediterranean Tree and Shrub Species: A New Approach to Quantify Climate-Growth Relationships

**J. Julio Camarero** [1,*] and **Álvaro Rubio-Cuadrado** [2]

1    Pyrenean Institute of Ecology (IPE-CSIC), 50192 Zaragoza, Spain
2    Departamento de Sistemas y Recursos Naturales, Escuela Técnica Superior de Ingeniería de Montes, Forestal y del Medio Natural, Universidad Politécnica de Madrid, Ciudad Universitaria s/n, 28040 Madrid, Spain; alvaro.rubio.cuadrado@upm.es
*    Correspondence: jjcamarero@ipe.csic.es; Tel.: +34-976-363-222 (ext. 880041)

**Abstract:** The quantification of climate–growth relationships is a fundamental step in tree-ring sciences. This allows the assessment of functional responses to climate warming, particularly in biodiversity and climate-change hotspots including the Mediterranean Basin. In this region, broadleaf tree and shrub species of pre-Mediterranean, subtropical origin, have to withstand increased aridification trends. However, they have not been widely studied to assess their long-term growth responses to climate and drought. Since these species evolved under less seasonal and wetter conditions than strictly Mediterranean species, we hypothesized that their growth would mainly respond to higher precipitation and water availability from spring to early summer. Here, we quantified climate–growth relationships in five of these broadleaf species showing different leaf phenology and wood type (*Pistacia terebinthus* L., *Pistacia lentiscus* L., *Arbutus unedo* L., *Celtis australis* L., and *Laurus nobilis* L.) by using dendrochronology. We calculated Pearson correlations between crossdated, indexed, mean ring width series of each species (chronologies) and monthly climate variables (mean temperature, total precipitation). We also calculated correlations between the species' chronologies and a drought index on 7-day scales. Lastly, we compared the correlation analyses with "*climwin*" analyses based on an information-theoretic approach and subjected to cross-validation and randomization tests. As expected, the growth of all species was enhanced in response to wet and cool conditions during spring and early summer. In some species (*P. lentiscus*, *A. unedo*, *C. australis*,) high prior-winter precipitation also enhanced growth. Growth of most species strongly responded to 9-month droughts and the correlations peaked from May to July, except in *L. nobilis* which showed moderate responses. The "*climwin*" analyses refined the correlation analyses by (i) showing the higher explanatory power of precipitation (30%) vs. temperature (7%) models, (ii) selecting the most influential climate windows with June as the median month, and (iii) providing significant support to the precipitation model in the case of *P. terebinthus* confirming that the radial growth of this species is a robust proxy of hydroclimate variability. We argue that "*climwin*" and similar frameworks based on information-theoretic approaches should be applied by dendroecologists to critically assess and quantify climate–growth relationships in woody plants with dendrochronological potential.

**Keywords:** *climwin*; dendrochronology; dendroclimatology; drought; *Arbutus unedo*; *Laurus nobilis*; Mediterranean; *Pistacia lentiscus*; *Pistacia terebinthus*; Standardized Precipitation-Evapotranspiration Index; tree rings

## 1. Introduction

The Mediterranean Basin is a biodiversity hotspot where many woody plant species are found across diverse habitats [1]. This diverse and endemism-rich vascular flora is menaced by several threats, including local exploitation of natural resources and habitat loss related to urbanization due to human pressure from residents or tourists, which depend on the shifting socio-economic development of each country, and regional climate warming [2]. The Mediterranean Basin is also considered a climate-change hotspot where current annual temperatures are ca. +1.5 °C higher than during the late 19th century, and such rapid warming may exacerbate drought stress [3]. Such climate trends make Mediterranean forests and scrublands vulnerable to climate change [4]. Therefore, we need better knowledge on how these woody species respond to climate in order to improve their conservation or management policies.

Quantifying radial growth through the study of tree rings allows a better understanding of functional responses of trees and shrubs to hydroclimate variability [5]. Specifically, the study of climate–growth relationships by relating tree-ring variables (usually ring width but also density, anatomy, isotopes, or chemical composition) with seasonal, monthly, weekly, or daily climate variables (air temperature, precipitation, water balance, radiation, etc.) has been widely used [5]. The rich Mediterranean woody flora provides a good opportunity to assess how the radial growth of tree and shrub species responds to climate by selecting those taxa with annual rings that can be crossdated [6–8].

In the Mediterranean Basin, most tree-ring studies have been developed in major forest species including conifers (pines, firs, cedars, junipers) and hardwood species (beech, oaks) [9,10]. Broadleaf tree and shrub species of pre-Mediterranean origin are understudied from a dendroecological point of view with some exceptions such as *Celtis australis* L. [11,12], *Pistacia lentiscus* L. [7,13] and *Arbustus unedo* L. [8,14–16]. The climate–growth relationships in other ecologically similar species forming annual rings including *Pistacia terebinthus* L. and *Laurus nobilis* L. have been little investigated to the best of our knowledge. These five hardwood, broadleaf tree and shrub species evolved under subtropical conditions before the typical dry-summer conditions appeared in the Mediterranean Basin during the second half of the Pliocene [17]. The studied species are considered descendants of old lineages that did not evolve under contemporary, Mediterranean climatic conditions characterized by a seasonal summer drought [18]. Such evolutionary origins impose constraints on the functioning of these species which display traits similar to subtropical species (e.g., big leaves) and evolved under climatic conditions characterized by a weaker seasonality than nowadays and scarce drought stress, similar to those currently experienced by the evergreen, broadleaved ("laurel") rainforests in Macaronesia [18]. The progressive drying of the Mediterranean Basin since the Miocene and the cooling and increasing seasonality during the Pleistocene and Pliocene, respectively, restricted the extent of subtropical species to refugia with warm and wet climate conditions [19].

Here we analyzed the climate- and drought–growth relationships in five species of pre-Mediterranean origin (*P. terebinthus*, *P. lentiscus*, *A. unedo*, *C. australis*, and *L. nobilis*). To achieve this aim we introduced the use of the R package "*climwin*" in tree-ring sciences, which allows the identification of climate windows on a response variable (tree radial growth in this case) based on information criteria and randomization approaches [20–22]. We argue that using this package in dendroecology could aid researchers to characterize climate–growth relationships, including the definition of climate windows of interests, more objectively than using correlative approaches. We compare this approach with the classical approach based on calculating correlation functions between ring-width series and climate variables [5]. Since the study species evolved under less seasonal and wetter conditions than nowadays, we expect them to show a positive growth response to water availability from spring to early summer. Under Mediterranean conditions radial growth usually peaks in spring and, under mild conditions (e.g., coastal sites), in autumn whereas wood formation is constrained by cold conditions in winter and by the dry summer [8,23,24]. Alternatively, in strictly Mediterranean species evolved during the late Pliocene under drier and more seasonal conditions

we would also expect some growth occurring in summer and responding to rare, wet conditions in that season.

## 2. Materials and Methods

### 2.1. Study Sites and Tree Species

We selected climatically limiting sites (e.g., xeric locations, open sites with steep slopes and/or shallow soils) where we expected the study trees to show year-to-year growth variability coupled with hydroclimate variability (Figure 1). The selected species are native to the Mediterranean Basin and three of them are evergreen species forming diffuse- to semi-ring-porous wood (*P. lentiscus*, *A. unedo*, *L. nobilis*), whilst the other two are deciduous species forming ring-porous wood (*P. terebinthus*, *C. australis*). All study species are trees except *P. lentiscus* which grows as a shrub.

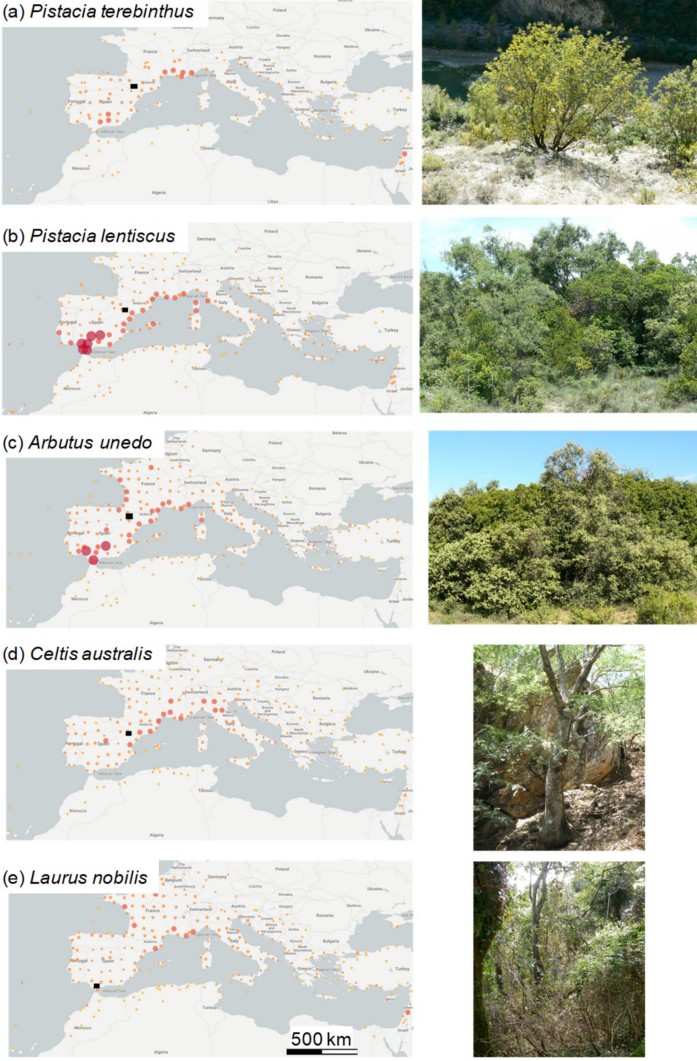

**Figure 1.** Maps show the distribution of the five study species in the Mediterranean Basin. Pictures show the studied tree species or sites where they were sampled. In the maps, the black squares indicate the approximate location of the study sites. Maps are based on occurrence data of the Global Biodiversity Information Facility database available from: https://www.gbif.org (accessed on 5 October 2020).

The *P. terebinthus* site (0°13′ E, 42°19′ N, 474 m) is located on a steep slope on rocky deposits, near the Cinca river. Soils are basic. The vegetation is a Mediterranean scrubland with scattered trees of oak species (*Quercus ilex* L. subsp. *ballota*, *Quercus faginea* Lam.) and shrub species (*Viburnum tinus* L.,

*Lonicera implexa* Ait.). The mean annual temperature is 11.4 °C and total annual precipitation is 660 mm with a summer drought lasting from July to August.

The *P. lentiscus* site (0°05′ E, 41°25′ N, 200 m) is located in a semi-arid region with a mean annual temperature of 15.9 °C and total annual precipitation of 313 mm [7]. In this area drought peaks in July and August but can extend from May to October. Soils are basic and develop over limestone and gypsum. The vegetation is dominated by shrub species (*Juniperus phoenicea* L., *Rhamnus lycioides* L., *Rosmarinus officinalis* L., *Genista scorpius* (L.) DC.) and Aleppo pine (*Pinus halepensis* Mill.).

The *A. unedo* site (0°47′ W, 42°18′ N, 750 m) is subjected to Mediterranean climate conditions. The vegetation is a mixed secondary forest with several tree and shrub species including *Q. ilex*, *Q. faginea*, *P. halepensis*, *Juniperus oxycedrus* L., *V. tinus*, and *P. lentiscus*. Soils are developed on clays with a bedrock of calcareous sandstone. The mean annual temperature is 11.3 °C and total annual precipitation is 635 mm [25].

The *C. australis* site is located in a canyon bottom (0°57′ W, 41°20′ N, 555 m) where temporal water courses provide some wetness within the study region subjected to semi-arid Mediterranean climate conditions. Mean annual temperature is 11.4 °C and annual precipitation is 428 mm [12]. As in the *P. lentiscus* sites, drought peaks from July to August. The geological substrate is limestone which leads to the formation of soils with shallow calcic horizon, karstic landscapes, and terrace gravels. The vegetation is dominated by *Hedera helix* L., *Acer monspessulanum* L. and *P. terebinthus*.

The *L. nobilis* site (5°35′ W, 36°30′ N, 560 m) is subjected to mild Mediterranean climate conditions with mean annual temperature of 16.7 °C, and annual precipitation of 880 mm. Summer drought usually lasts from June to September. Soils are acid and well-developed with sandy and clay textures. The vegetation is dominated by other tree (*Ilex aquifolium* L., *Frangula alnus* subsp *baetica* (Rev. and Willk.) Rivas Goday ex Devesa, *Alnus glutinosa* (L.) Gaertn., *Quercus canariensis* Willd., *Quercus suber* L., *A. unedo*) and shrub species (*Rhododendron ponticum* subsp. *baeticum* (Boiss. and Reuter) Hand.-Mazz., *V. tinus*).

In the five study sites the wettest seasons are winter, spring, and autumn. In all sites, particularly in the *P. lentiscus* and *C. australis* semi-arid sites, winter is drier than spring, with a precipitation peak in May, excepting in the *L. nobilis* site where more precipitation is recorded in winter than in spring. None of the selected sites had been affected by local disturbances like forest thinning, overgrazing, or wildfires for the past 40 years. In the *P. lentiscus* site, a dieback episode related to the severe 2012 drought mainly affected *J. phoenicea* which presented a mean crown defoliation rate of 66% but *P. lentiscus* was little affected with 8% defoliation rate [7].

## 2.2. Climate Data and Drought Index

At all sites climatic data (mean air temperature, precipitation) and drought indices were obtained from a ~1.1 km$^2$-gridded and homogenized database of climate records in Spain for the period 1962–2015 [26]. We used monthly climate data and 7-day drought indices. We selected the Standardized Precipitation-Evapotranspiration Index (SPEI) to quantify drought severity at several time resolutions [27], with dry conditions corresponding to negative SPEI values. To evaluate the time and duration when drought most affected growth, we calculated SPEI values at 7-day resolution considering 1, 3, 6, 9, 12, 24 and 36 monthly periods. We plotted Pearson correlations based on 3- (SPEI3), 6- (SPEI6) and 9-month (SPEI9) SPEI values following previous studies in other Mediterranean species which showed these were the most relevant periods [28,29]. Then, we compared these results with the time windows and monthly periods based on "*climwin*" analyses.

## 2.3. Field Sampling and Processing of Tree-Ring-Width Data

We selected between 12 and 24 adult, dominant, apparently healthy individuals of each species. We measured the basal diameter and height of the sampled individuals using tapes and clinometers (Suunto, Vantaa, Finland), respectively (Table 1). Sampled *P. lentiscus* showed an average height of 1.3 m, whilst the height of the other species ranged from 6.0–7.5 m in *P. terebinthus*, *A. unedo*, and *L. nobilis* to 11.0 m in *C. australis*. In the case of *P. terebinthus* and *P. lentiscus* we obtained cross-sections using a

hand saw because most individuals showed stems with diameters lower than 15 cm (Table 1). We also sampled five sections for *A. unedo* to check the crossdating of cores since this species may form wedging rings [8]. In the other three species, cores were extracted at 1.3 m and perpendicular to the stem using Pressler increment borers (Haglöf, Långsele, Sweden). The wood samples were carefully sanded until rings were visible, and visual crossdating was performed. The crossdating was checked using the COFECHA program [30]. Two radii per individual were crossdated, and ring widths were measured to the nearest 0.001 mm using a Lintab-TSAP measuring device (RinntechTM, Heidelberg, Germany).

**Table 1.** Study species, basal diameter of sampled individuals, and ring-width statistics. Values are means ± standard deviations for the common period 1984–2015. Abbreviations: AC1, first-order autocorrelation; MS, mean sensitivity; rbar, mean correlation between indexed ring-width series. The last column indicates related bibliographic references studying the same species or sites. The best replicated period was defined as that with EPS ≥ 0.85, where EPS is the expressed population signal.

| Species | Diameter (cm) | No. Individuals | No. Radii | Best Replicated Period | Mean Ring Width (mm) | AC1 | MS | Rbar | Related Reference |
|---|---|---|---|---|---|---|---|---|---|
| *Pistacia terebinthus* | 15.7 ± 1.1 | 20 | 40 | 1973–2015 | 1.06 ± 0.51 | 0.49 | 0.26 | 0.54 | - |
| *Pistacia lentiscus* | 12.0 ± 0.9 | 14 | 28 | 1984–2015 | 0.65 ± 0.35 | 0.28 | 0.39 | 0.59 | [7,13] |
| *Arbutus unedo* | 20.2 ± 1.8 | 12 | 24 | 1980–2015 | 0.93 ± 0.63 | 0.45 | 0.41 | 0.46 | [8,14–16] |
| *Celtis australis* | 27.2 ± 3.8 | 12 | 24 | 1961–2015 | 0.85 ± 0.21 | 0.62 | 0.27 | 0.55 | [11,12] |
| *Laurus nobilis* | 22.0 ± 2.1 | 24 | 48 | 1971–2015 | 2.37 ± 1.04 | 0.46 | 0.21 | 0.40 | - |

The series of tree-ring-width data were detrended and standardized using the software ARSTAN v.44 [31]. Detrending allows the removal of growth trends due to changes in size and age of trees and shrubs. First, negative exponential curves were fitted to ring-width data and residuals were obtained by dividing observed from fitted values. Second, the resulting detrended series of ring-width indices were pre-whitened with 1st-order autoregressive models to remove year-to-year growth persistence. Individual ring-width indices were combined into mean series for each species (ring-width chronologies) using a biweight robust mean [5]. To characterize the species' chronologies we calculated the mean, standard deviation, and first-order autocorrelation of raw ring-width data, and the mean sensitivity (MS) and mean correlation between indexed ring-width series for the common period 1984–2015 [32]. The MS measures the relative change in width indices between consecutive rings with higher values corresponding to higher growth variability [5]. Lastly, we calculated Pearson correlations between the species' chronologies to assess their coherence in year-to-year growth variability. We considered an expressed population signal (EPS) ≥ 0.85 threshold to assess the adequacy of replication in the species' chronologies [33].

*2.4. Climate– and Drought–Growth Relationships*

To quantify how climate (temperature, precipitation) is related to interannual growth variability we calculated Pearson correlations between species ring-width chronologies and monthly climate data (mean temperature, total precipitation) from the previous to current (growth year) September, following previous studies [7,9,10,12]. We present correlations with precipitation because they showed similar results as the correlations with the climatic water balance, i.e., the difference between precipitation and potential evapotranspiration calculated using the Penman–Monteith Equation [34].

To characterize drought-growth relationships we calculated Pearson correlations between the chronologies and 7-day series of SPEI3, SPEI6, and SPEI9 and plotted them. We selected 7-day-long series because their resolution is short enough to reflect changes in intra-annual growth dynamics and the resulting responses to water shortage [24,35]. In these analyses we showed the 0.05 and 0.01 significance thresholds.

*2.5. Using Climwin to Define the Climate Window Most Tightly Related to Growth*

We used the "*climwin*" package in R v 3.6.2. to identify the best climate predictors of growth and to select the most relevant climate window [20,21]. Developers of the package encourage users to consider a wide range of windows and first, we tested an annual time window despite phenological, dendroecological, and xylogenesis studies indicating that radial growth peaks in spring in the five study species, thus suggesting seasonal or monthly windows [11–16]. This package uses an information-theoretic approach to select the most parsimonious model by minimizing the corrected Akaike information criterion (AICc) [36]. In "*climwin*" the best supported model has the lowest ΔAICc compared to the null model without climate variables. The most relevant statistics provided by the "*climwin*" output are the ΔAICc of the best model, the timing of the best climate window (when the window opens and closes), and the beta-estimates, i.e., the coefficient estimates of the effect of the climate signal on growth (slopes of the linear models). Randomization tests are calculated to determine the likelihood of a climate signal, i.e., the expected distribution of ΔAICc values in a data set where no response to climate exists. To obtain unbiased and reliable parameter estimates and model statistics derived from our best model (e.g., $R^2$, slope, window duration) "*climwin*" performs a K-fold cross-validation by dividing the dataset into training and test datasets. The package produces several output panels showing: (i) how ΔAICc changes as a function of the fitted climate windows, and strongly supported windows are often plotted together in that panel; (ii) the spread of model coefficients across all fitted climate windows, and (iii) the histogram of ΔAICc values from all models run on the randomized sets and the ΔAICc of the best model fitted on the observed data. In this histogram the percentage of randomizations that generates a ΔAICc value that is at least as low as the ΔAICc value of the best model (the statistic $P_{\Delta AICc}$) is a measure of how likely it is that one obtained the observed ΔAICc by chance. The package also produces another panel plotting Akaike model weights with higher values, indicating a greater confidence that the selected model is the true best model [37]. In the gridded panels of ΔAICc, Akaike weights and beta estimates, the *x* and *y* axes indicate the times (days, weeks, months, etc.) when climate windows open and close, respectively. Another relevant panel shows the opening and closing points of the time windows encompassing models best supported by the data. Finally, the last plotted panel shows the relationship between the climate signal from the selected time window and the biological response (here the ring-width index).

We assessed the responses to temperature, precipitation, and SPEI. The response function was considered linear and we used an absolute time window assuming relatively small variation among years in the timing of the growth response. This last assumption may not always be appropriate in Mediterranean species which may show bimodal and unimodal growth patterns in different sites and during climatically contrasting years [24,35,38]. We compared the results applying and not applying the randomization tests and the cross-validation, using in this case 20 training and 20 test datasets.

## 3. Results

*3.1. Tree-Ring Statistics of Species' Mean Chronologies*

The shortest chronology (period 1984–2015) corresponded to the shrub *P. lentiscus*, which presented the smallest ring widths (mean = 0.65 mm). The widest rings were formed by *L. nobilis* (mean = 2.37 mm). The first-order autocorrelation (AC1) was maximum (0.62) in *C. australis* indicating a higher year-to-year growth similarity, whilst the lowest AC1 was observed in P. lentiscus (0.28). The mean sensitivity (MS) peaked in *A. unedo* (0.41) and *P. lentiscus* (0.39), and showed minimum values in *L. nobilis* (0.21), suggesting a lower year-to-year growth variability in this species and a lower responsiveness to climate. All species' chronologies showed an EPS ≥ 0.85 for the common period 1984–2015 and, therefore, they were considered well-replicated. The coherence between individual series, measured through the mean correlation between indexed ring-width series (rbar), was the highest in *P. lentiscus* (0.59), *C. australis* (0.55), and *P. terebinthus* (0.54), intermediate in *A. unedo* (0.46), and the lowest in *L. nobilis* (0.40).

The species' chronologies showed some common patterns as pronounced growth drops in dry years such as 1993–1995, 2005, and 2012 and growth increments in wet years such 1988, 1997, and 2003 (Figure 2). The similarity between species' chronologies was quantified through Pearson correlations and showed that all were significantly ($p < 0.05$) and positively correlated among them, excepting the *L. nobilis* chronology, which only showed a significant positive correlation with the *P. terebinthus* chronology (Table 2). The association between the *L. nobilis* and *P. terebinthus* chronologies is remarkable given that these two study sites are separated by a linear distance of 810 km.

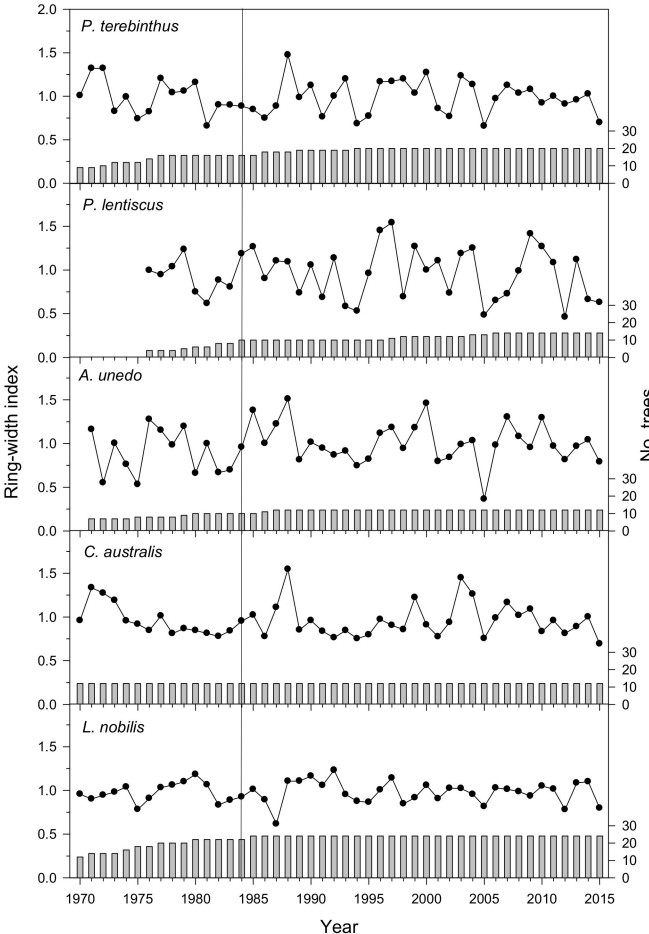

**Figure 2.** Indexed, mean ring-width series (chronologies) of the five study species. The bars show the sample size (right *y* axis), i.e., the number of individuals measured each year. The vertical line indicates the best-replicated, common period (1984–2015).

**Table 2.** Pearson correlations (values below the diagonal) and associated significance levels (values above the diagonal) calculated for the common period 1984–2015 between the chronologies of the five study species. Significance levels: * $p < 0.05$; ** $p < 0.01$; *** $p < 0.001$.

| | *Arbutus unedo* | *Celtis australis* | *Pistacia terebinthus* | *Pistacia lentiscus* | *Laurus nobilis* |
|---|---|---|---|---|---|
| *Arbutus unedo* | | 0.001 | 0.006 | 0.001 | 0.164 |
| *Celtis australis* | 0.526 *** | | 0.001 | 0.020 | 0.625 |
| *Pistacia terebinthus* | 0.484 ** | 0.580 *** | | 0.022 | 0.021 |
| *Pistacia lentiscus* | 0.518 *** | 0.372 * | 0.411 * | | 0.204 |
| *Laurus nobilis* | 0.234 | 0.081 | 0.414 * | 0.208 | |

The relationship between the linear distance between sites and the correlations between site ring-width chronologies was significant and negative ($r = -0.74$, $p = 0.01$, $n = 10$ pairs), illustrating a loss of growth synchrony among species as site-to-site distance increased. However, if the *L. nobilis*

site was not considered, the correlation turned positive ($r = 0.77$, $p = 0.07$, $n = 6$ pairs of chronologies), suggesting a common growth signal among the rest of species' chronologies which all corresponded to sites located in northeastern Spain.

### 3.2. Climate– and Drought–Growth Relationships Based on Pearson Correlations

The correlation functions showed that growth of most tree species was improved in response to wet and cool spring to early summer conditions (Figure 3). In *P. terebinthus*, high precipitation from April to June and cool June to July conditions enhanced growth. In *P. lentiscus*, growth also increased in response to high precipitation in the current April, June, and July, but the highest response was found in response to high precipitation in the previous winter (December and January), i.e., before the start of ring formation. Warm previous-November and current-April conditions also showed positive correlations with ring-width indices, but warm July conditions, probably associated with drought stress, reduced growth. In *A. unedo*, growth increased as April and June precipitation and September temperature did, but decreased in response to July high temperatures. In *C. australis*, a higher precipitation in the previous December and also in the current March to May period was associated with improved growth, and warm February and March conditions constrained growth. Lastly, in *L. nobilis* growth increased in response to high precipitation during the previous October and December and in the current June, and also in response to warm previous-December and current-May conditions.

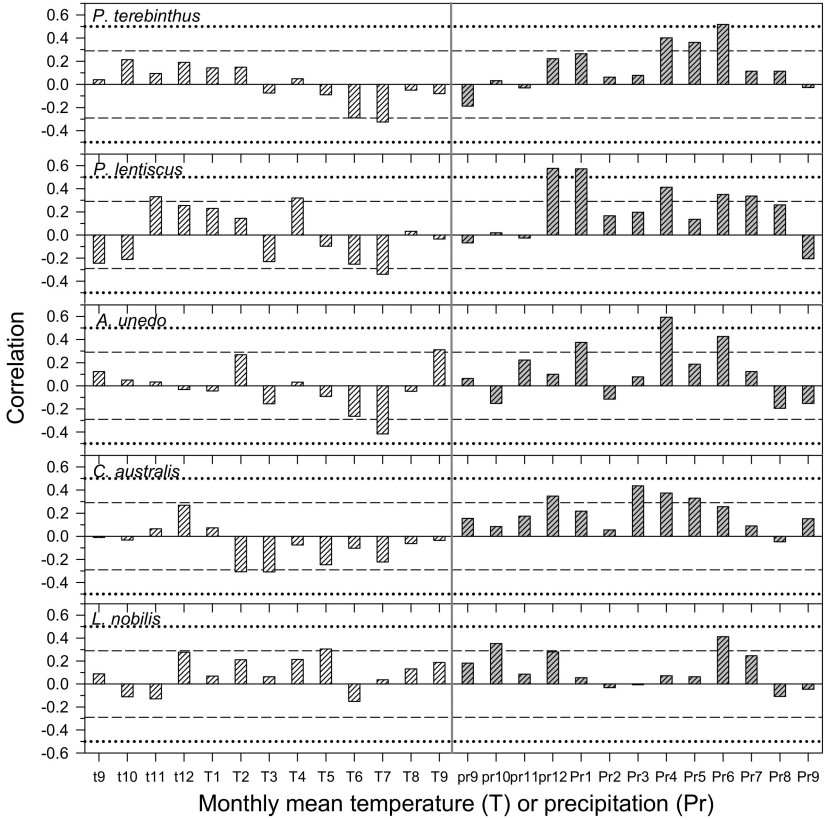

**Figure 3.** Pearson correlations between monthly climate variables (*x* axis) and mean series of ring-width indices of the five study species. The climate variables (T, temperature, white bars; Pr, precipitation, grey bars) are abbreviated by lowercase and uppercase letters corresponding to the previous and current years, respectively. Current year is the year of ring formation. Numbers indicate months. Dashed and dotted horizontal lines correspond to the 0.05 and 0.01 significance levels, respectively.

The correlations calculated with 7-day SPEI values showed a higher responsiveness to drought in *P. terebinthus* and *P. lentiscus*, intermediate responsiveness in *A. unedo* and *C. australis*, and a

lower responsiveness in *L. nobilis* (Figure 4). In *P. terebinthus*, the highest correlations (*r* = 0.74) were found for 9-month SPEI values (SPEI9) from mid-to-late June (days 170 to 180). In *P. lentiscus*, the highest correlations (*r* = 0.76) were also found for SPEI9 during June (days 160 to 180). In *A. unedo*, SPEI-growth correlations peaked (*r* = 0.68) for 6-month SPEI values (SPEI6) in July (days 180 to 210). In *C. australis*, correlations peaked (*r* = 0.66) again for SPEI9 in mid-to-late May (day 140). In *L. nobilis*, maximum correlations (*r* = 0.40) were observed for SPEI9 between April and late June (days 100 to 180).

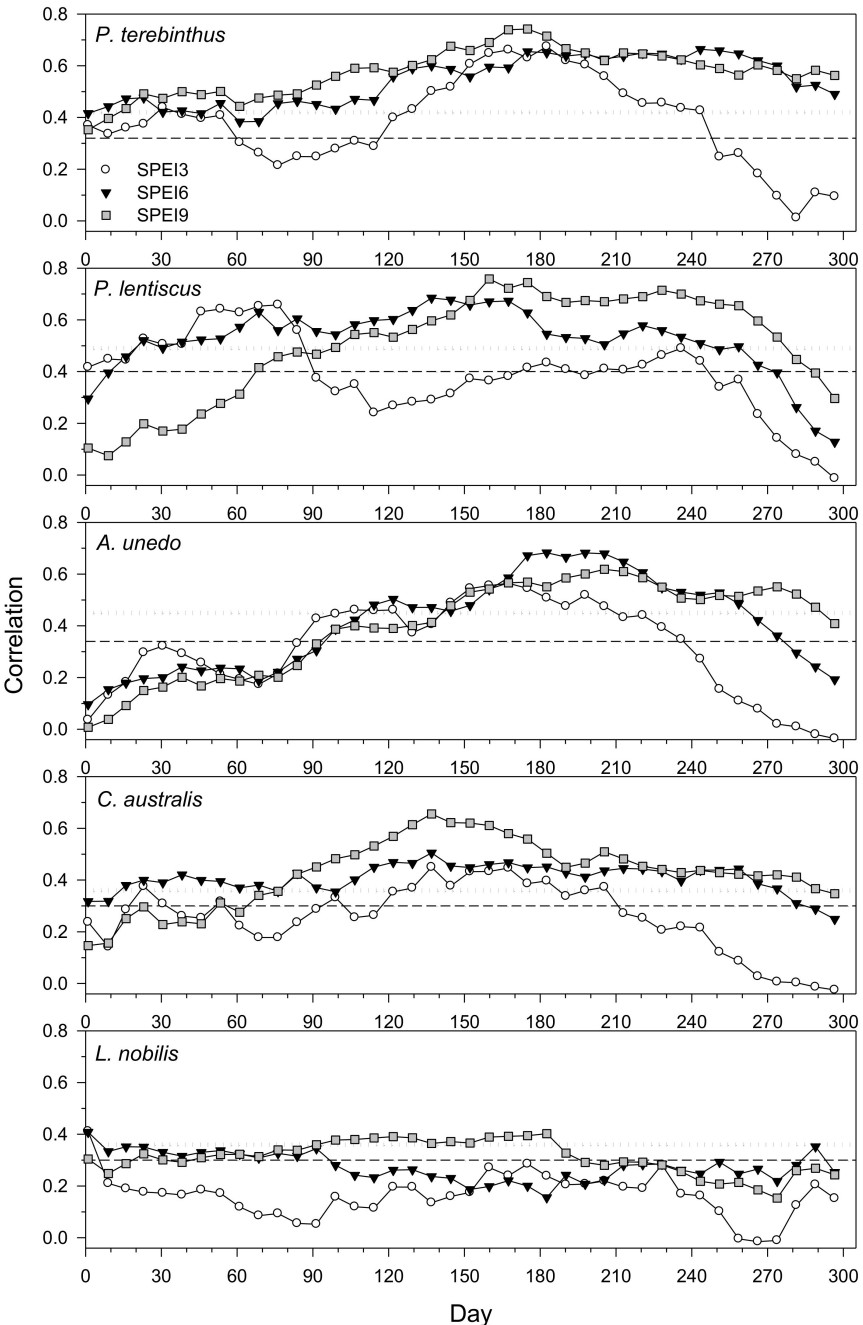

**Figure 4.** Pearson correlations between 7-day SPEI data (*x* axis, days) and mean series of ring-width indices (chronologies) of the five study species. The SPEI drought index was calculated for temporal resolutions of 3 (SPEI3), 6 (SPEI6), and 9 (SPEI9) months. Dashed and dotted horizontal lines correspond to the 0.05 and 0.01 significance levels, respectively.

### 3.3. Climate–Growth Relationships According to "Climwin" Analyses

The climate–growth relationships analyzed with "*climwin*" are summarized in Table 3 and the selected models are presented in Table S1. The linear models found significant associations in all species and the two climate variables considered (temperature, precipitation), excepting temperature in the case of *P. lentiscus*. The maximum amount of growth variability ($R^2$) explained by climate corresponded to January to August precipitation in the case of *P. lentiscus* ($R^2 = 0.508$; Figure 5) followed by February to July precipitation in the case of *A. unedo* ($R^2 = 0.437$), and April to June precipitation in the case of *P. terebinthus* ($R^2 = 0.393$; Figure 6). In the case of temperature, the explained growth variance was much lower with maximum values in the case of *A. unedo* ($R^2 = 0.163$; June–July temperature) and *L. nobilis* ($R^2 = 0.140$; February–May temperature). However, when we assessed the significance of fitted linear models with K-fold cross-validation and randomization, only precipitation in the case of *P. terebinthus* remained significant, explaining 36.2% of growth variability (see the histogram in Figure 6b). In the precipitation models explaining more than 25% of growth variability, most of the selected climate windows encompassed spring and summer months (*P. lentiscus*, *P. terebinthus*, *A. unedo*), but also previous-winter months (*P. lentiscus*, *A. unedo*, *C. australis*). After cross-validation and randomization, the climate windows remained similar in the precipitation models with higher amounts of explained growth variance in the case of *P. terebinthus* (April–July), *P. lentiscus* (January–August), and *A. unedo* (March–July). The amount of growth variance explained by temperature in the *A. unedo* and *L. nobilis* models was very low (0.4–1.3%) after cross-validation and randomization, and it was also reduced in the *L. nobilis* temperature model (1.8%).

### 3.4. Drought–Growth Relationships According to "Climwin" Analyses

The drought–growth relationships analyzed with "*climwin*" and using the 7-day SPEI data at several monthly resolutions are summarized in Table 4 and Figure 7. The coefficients of the two SPEI models with the lowest $\Delta$AICc for each species are shown in Table S2. The "*climwin*" analyses showed significant relationships between the species' chronologies and the SPEI for all species excepting *L. nobilis* (Table 4). The highest $R^2$ values corresponded to 9-month-long droughts in the case of *P. lentiscus* ($R^2 = 0.575$), *P. terebinthus* ($R^2 = 0.547$), and *C. australis* ($R^2 = 0.430$), and to 1-month-long ($R^2 = 0.488$) and 6-month-long ($R^2 = 0.469$) droughts in the case of *A. unedo* (Figure 7). The climate windows differed among species, but the most frequently observed window for the most responsive species (*P. lentiscus*, *P. terebinthus*) included the growing-season period from early May to mid August. The mean climate window considering the SPEI9 models, which presented the lowest $\Delta$AICc and highest $R^2$ values (Table 4), spanned days 112 to 169, i.e., from 22 April to 18 June.

**Table 3.** Summary of the climate–growth relationships based on "*climwin*" analyses. In all cases linear models and mean values of the climate variables were used. Significant ($p < 0.05$) $R^2$ values are shown in bold characters. AICc: Akaike information criterion.

| Species | Climate Variable | Linear Model | | | | Linear Model Using K-Fold Cross-Validation and Randomization Method | | | |
|---|---|---|---|---|---|---|---|---|---|
| | | Climate Window | ΔAICc | $R^2$ | $p$ | Climate Window | ΔAICc | $R^2$ | $p$ of the Randomization |
| *P. terebinthus* | Temperature | June–July | −4.57 | **0.136** | 0.0108 | July | −2.40 | 0.136 | 0.4468 |
| | Precipitation | April–June | −21.15 | **0.393** | 0.0001 | April–July | −16.73 | **0.362** | 0.0008 |
| *P. lentiscus* | Temperature | June–July | 0.47 | 0.082 | 0.1664 | May–July | −2.29 | 0.0685 | 0.4976 |
| | Precipitation | January–August | −15.14 | **0.508** | 0.0001 | January–August | −2.75 | **0.508** | 0.4378 |
| *A. unedo* | Temperature | June–July | −3.64 | **0.163** | 0.0178 | June | −0.84 | 0.013 | 0.7376 |
| | Precipitation | February–July | −17.13 | **0.437** | 0.0001 | March–July | −8.36 | 0.427 | 0.0510 |
| *C. australis* | Temperature | February–March | −6.41 | **0.124** | 0.0040 | February–March | −2.72 | 0.124 | 0.3502 |
| | Precipitation | January–May | −19.87 | **0.288** | 0.0001 | January–December | −4.10 | 0.197 | 0.2730 |
| *L. nobilis* | Temperature | February–May | −3.68 | **0.140** | 0.0174 | October | −2.36 | 0.004 | 0.4038 |
| | Precipitation | June–July | −6.86 | **0.206** | 0.0033 | April–October | −2.49 | 0.018 | 0.5224 |

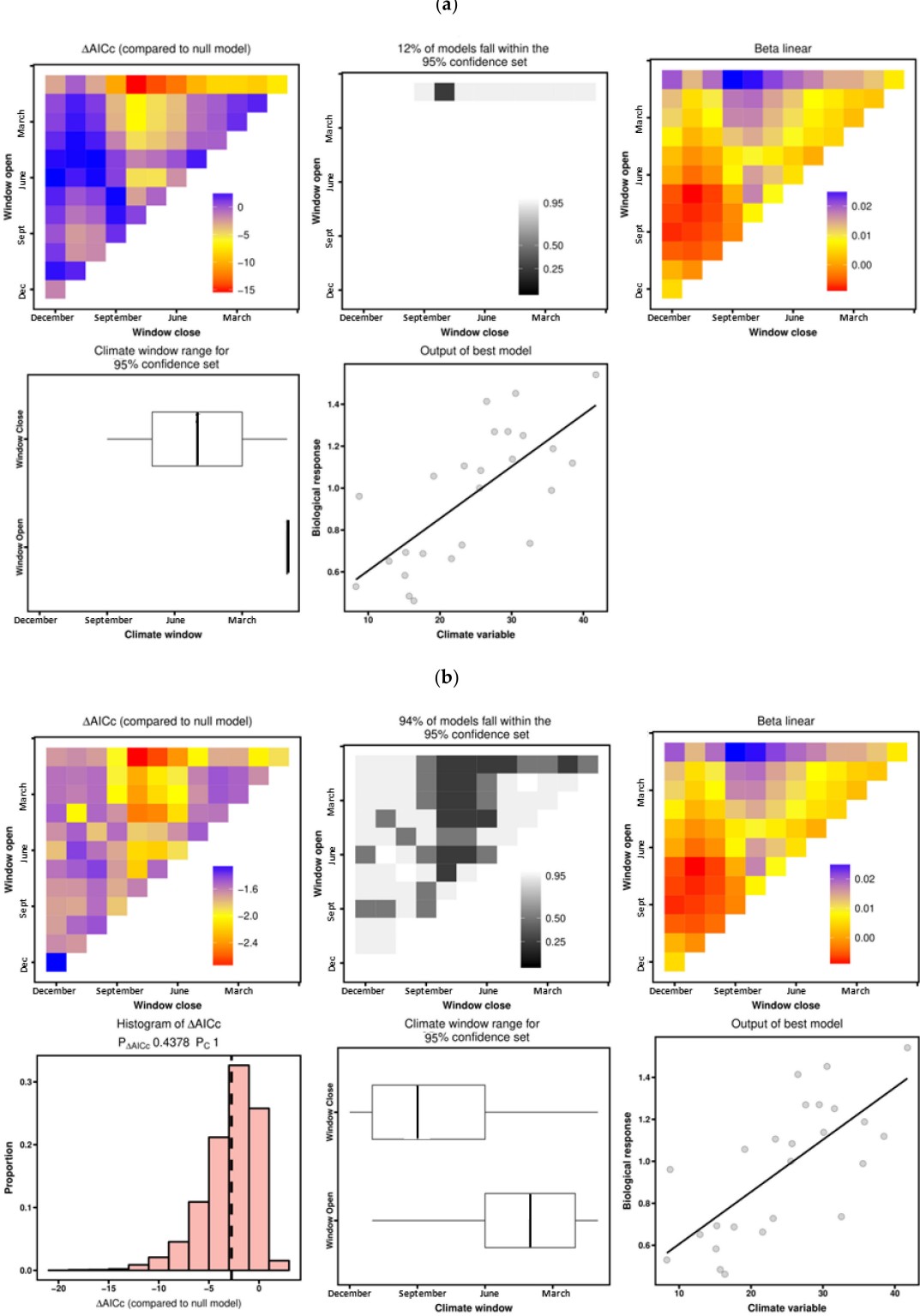

**Figure 5.** Examples of "*climwin*" output panels by calculating (**a**) linear models without and (**b**) with K-fold cross-validation and randomization method. The example corresponds to precipitation and the *Pistacia lentiscus* chronology. The model was not significant ($p < 0.05$) according to randomization tests (histogram of ΔAICc values; the vertical dashed line shows the ΔAICc of the best model fitted on the observed data).

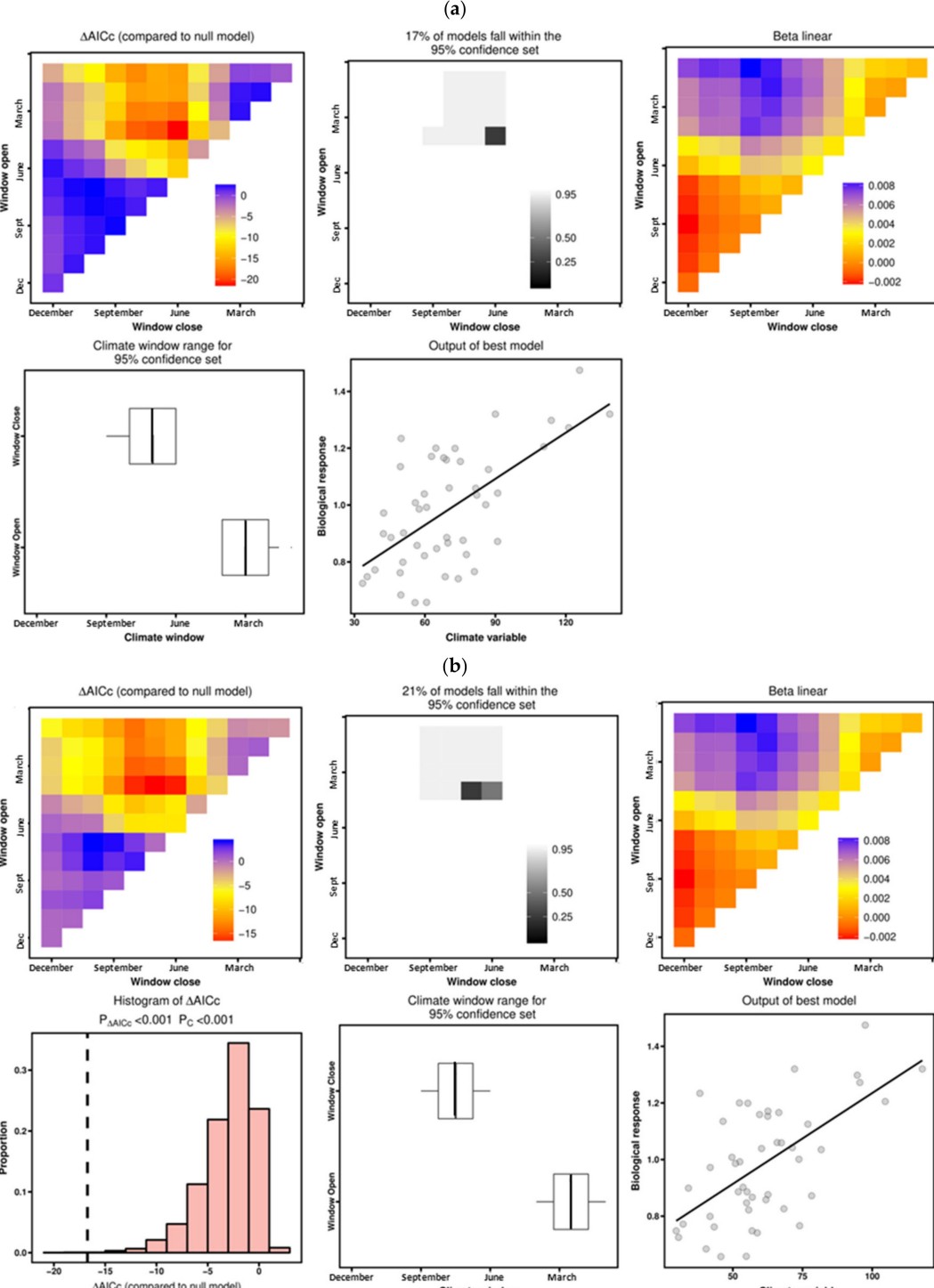

**Figure 6.** Examples of "*climwin*" output panels by calculating (**a**) linear models without and (**b**) with K-fold cross-validation and randomization method. The example corresponds to precipitation and the *Pistacia terebinthus* chronology. The model was significant ($p < 0.05$) according to randomization tests (histogram of ΔAICc values; the vertical dashed line shows the ΔAICc of the best model fitted on the observed data).

**Table 4.** Summary of the drought–growth relationships based on *"climwin"* analyses and using the SPEI 7-day resolution data calculated at 1, 3, 6, 9, 12, 24, and 36 monthly periods. In all cases linear models and mean SPEI values were used. Significant ($p < 0.05$) $R^2$ values are shown in bold characters. The abbreviation "DOY" stands for Julian day of the year.

| Species | SPEI Period (Months) | Climate Window (DOY) | ΔAICc | $R^2$ | $p$ | $p$ of the Randomization |
|---|---|---|---|---|---|---|
| *P. terebinthus* | 1 | 86–204 | −23.37 | **0.501** | $9.51 \times 10^{-7}$ | <0.001 |
| | 3 | 100–239 | −22.40 | **0.488** | $1.52 \times 10^{-8}$ | <0.001 |
| | 6 | 2–330 | −24.21 | **0.513** | $6.31 \times 10^{-7}$ | <0.001 |
| | 9 | 156–169 | −26.95 | **0.547** | $1.68 \times 10^{-7}$ | <0.001 |
| | 12 | 121–330 | −26.76 | **0.545** | $1.84 \times 10^{-7}$ | <0.001 |
| | 24 | 324–330 | −16.24 | **0.395** | $3.08 \times 10^{-8}$ | <0.001 |
| | 36 | 233–267 | −6.14 | **0.206** | $4.82 \times 10^{-3}$ | 0.012 |
| *P. lentiscus* | 1 | 2–232 | −10.65 | **0.411** | $5.51 \times 10^{-4}$ | 0.042 |
| | 3 | 2–239 | −13.19 | **0.468** | $1.62 \times 10^{-4}$ | 0.005 |
| | 6 | 2–232 | −17.62 | **0.555** | $1.96 \times 10^{-9}$ | 0.001 |
| | 9 | 149–155 | −18.80 | **0.575** | $1.12 \times 10^{-9}$ | <0.001 |
| | 12 | 212–218 | −17.49 | **0.552** | $2.09 \times 10^{-8}$ | <0.001 |
| | 24 | 240–246 | −1.96 | 0.166 | $4.29 \times 10^{-2}$ | 0.175 |
| | 36 | 233–239 | −0.04 | 0.100 | $1.23 \times 10^{-1}$ | 0.327 |
| *A. unedo* | 1 | 16–218 | −20.33 | **0.488** | $4.37 \times 10^{-7}$ | <0.001 |
| | 3 | 65–239 | −17.24 | **0.439** | $1.96 \times 10^{-9}$ | <0.001 |
| | 6 | 163–211 | −19.09 | **0.469** | $7.99 \times 10^{-8}$ | 0.001 |
| | 9 | 198–204 | −13.98 | **0.383** | $9.68 \times 10^{-7}$ | 0.002 |
| | 12 | 191–218 | −10.49 | **0.316** | $5.44 \times 10^{-4}$ | 0.007 |
| | 24 | 331–337 | −2.43 | 0.133 | $3.41 \times 10^{-2}$ | 0.135 |
| | 36 | 205–211 | 0.66 | 0.050 | $2.03 \times 10^{-1}$ | 0.467 |
| *C.australis* | 1 | 86–155 | −7.66 | 0.199 | $2.15 \times 10^{-3}$ | 0.11 |
| | 3 | 2–204 | −9.48 | **0.230** | $8.50 \times 10^{-4}$ | 0.023 |
| | 6 | 2–246 | −14.85 | **0.317** | $5.68 \times 10^{-5}$ | 0.001 |
| | 9 | 128–134 | −22.98 | **0.430** | $1.02 \times 10^{-6}$ | <0.001 |
| | 12 | 205–211 | −17.39 | **0.354** | $1.60 \times 10^{-5}$ | <0.001 |
| | 24 | 338–344 | −5.64 | **0.162** | $6.18 \times 10^{-3}$ | 0.026 |
| | 36 | 275–281 | −1.72 | 0.085 | $5.13 \times 10^{-2}$ | 0.135 |
| *L. nobilis* | 1 | 282–288 | −2.95 | 0.142 | $2.59 \times 10^{-2}$ | 0.609 |
| | 3 | 338–351 | −0.88 | 0.089 | $8.11 \times 10^{-2}$ | 0.726 |
| | 6 | 317–330 | −1.66 | 0.109 | $5.22 \times 10^{-2}$ | 0.446 |
| | 9 | 338–351 | −0.20 | 0.072 | $1.20 \times 10^{-1}$ | 0.568 |
| | 12 | 2–8 | −1.05 | 0.094 | $7.34 \times 10^{-2}$ | 0.253 |
| | 24 | 338–351 | 0.74 | 0.046 | $2.14 \times 10^{-1}$ | 0.478 |
| | 36 | 282–288 | 1.41 | 0.028 | $3.37 \times 10^{-1}$ | 0.629 |

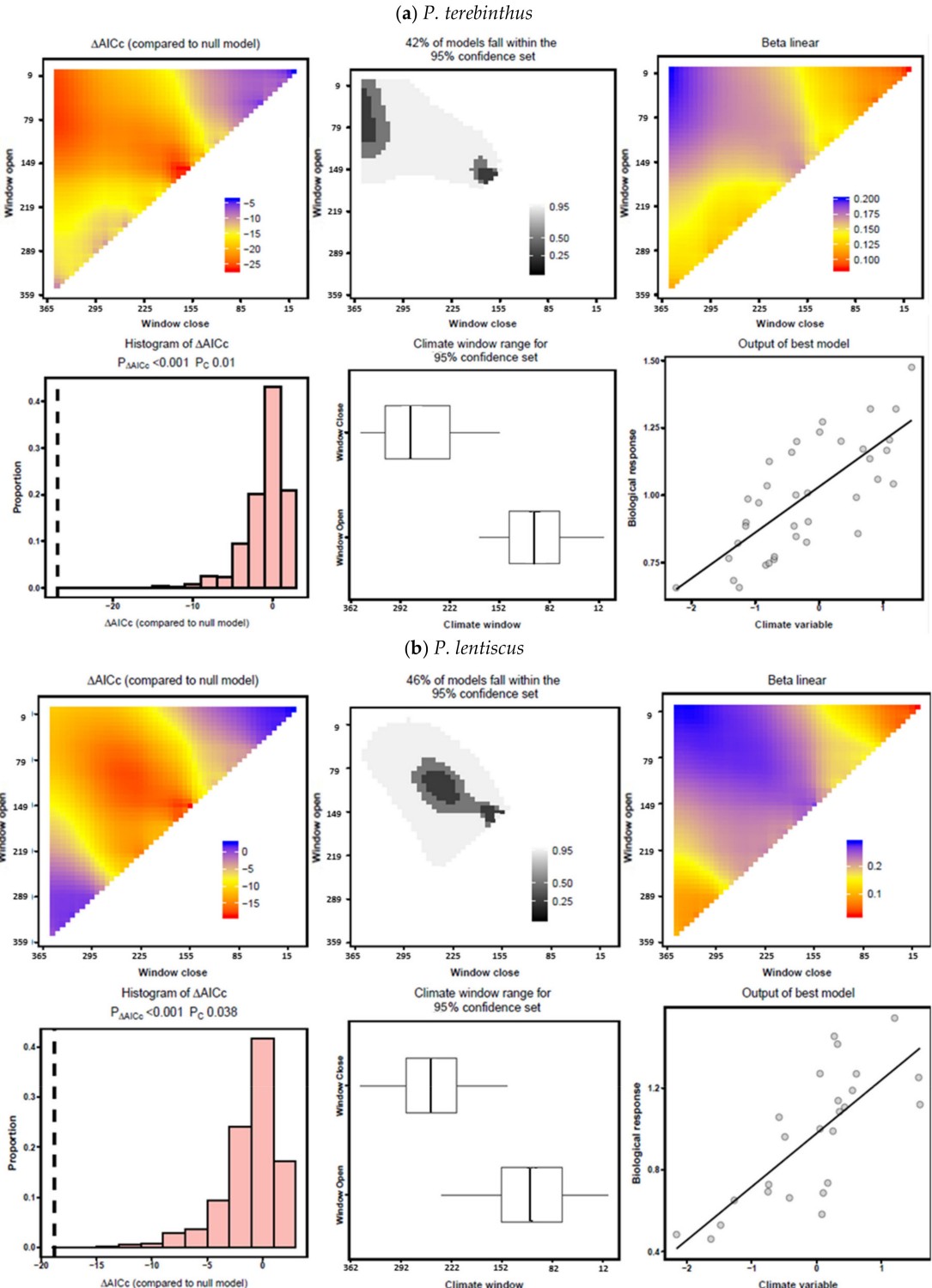

**Figure 7.** *Cont.*

(**c**) *A. unedo*

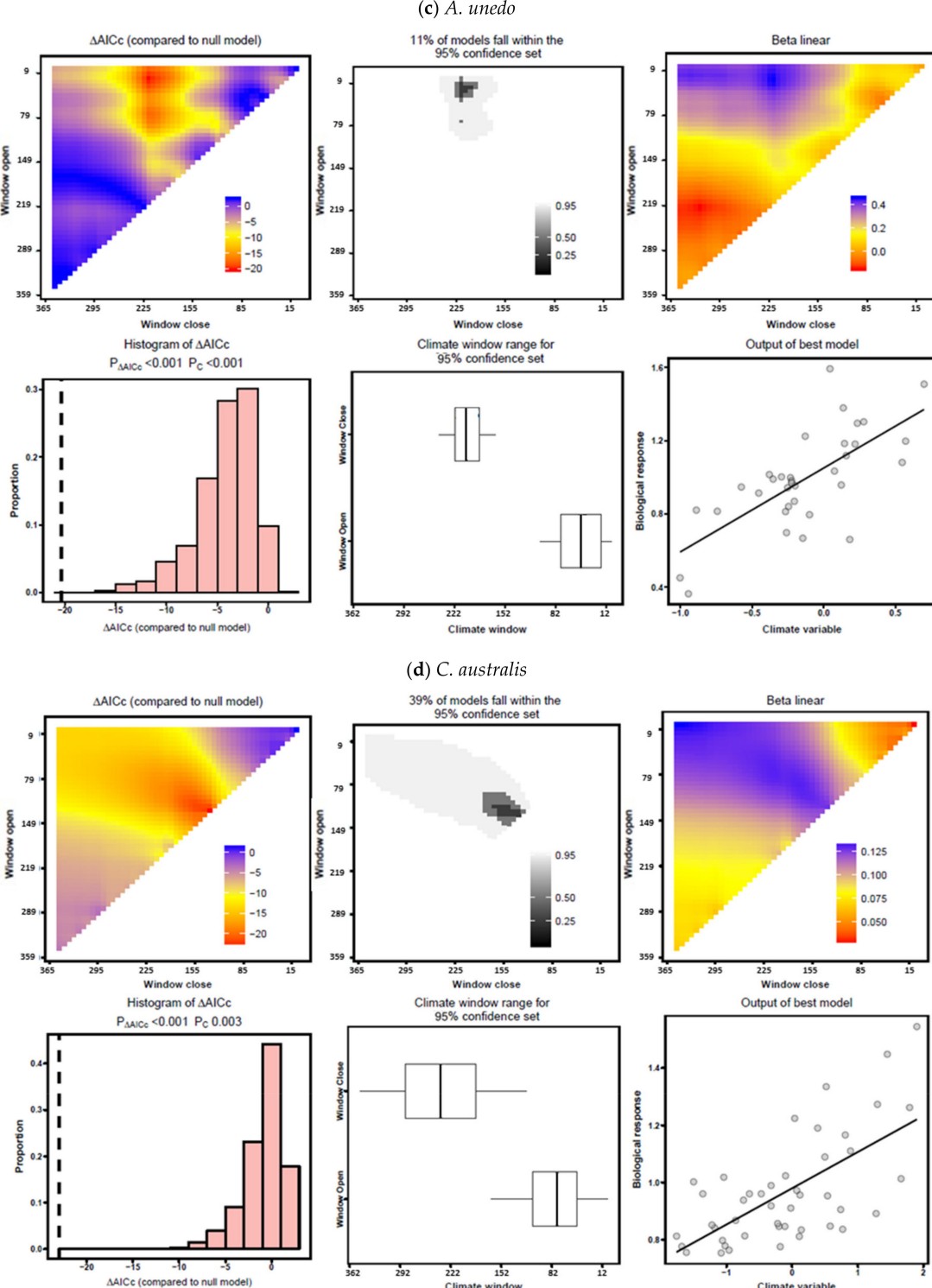

(**d**) *C. australis*

**Figure 7.** *Cont.*

(**e**) *L. nobilis*

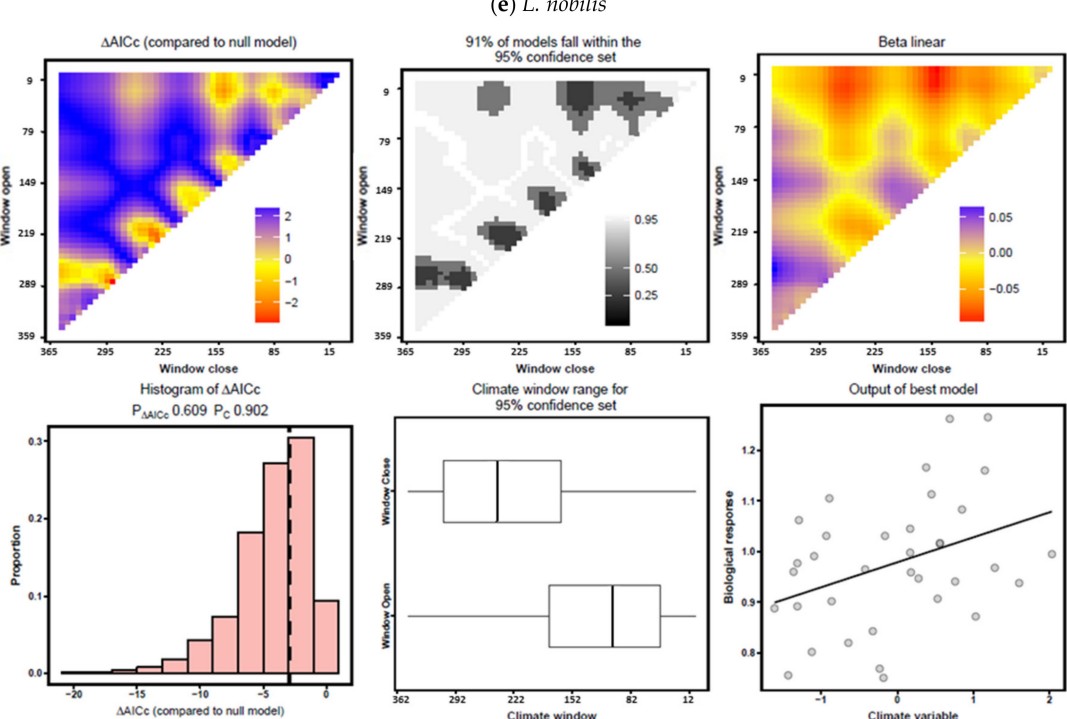

**Figure 7.** "*Climwin*" analyses showing associations between growth variability and the SPEI drought index. The plots correspond to the selected SPEI monthly values based on the models showing the highest $R^2$ and lowest ΔAICc values (see Table 4): 9-month SPEI values in the case of *P. terebinthus* (**a**), *P. lentiscus* (**b**), and *C. australis* (**d**), and 1-month SPEI values in the case of *A. unedo* (**c**) and *L. nobilis* (**e**). Windows open and close values are the Julian day of the year (DOY).

## 4. Discussion

We found a positive growth response in the five Mediterranean tree and shrub species of pre-Mediterranean origin (*P. terebinthus*, *P. lentiscus*, *A. unedo*, *C. australis,* and *L. nobilis*) to water availability from spring to early summer (Figure 3; Figure 4), when the growth rates peak. This finding agrees with previous dendroecological studies in some of these species as *P. lentiscus* [7,13], *A. unedo* [8,14–16], and *C. australis* [11,12]. In the case of *L. nobilis*, only a local study was found reporting a positive response to summer rainfall in a wet, oceanic site located in NW Spain [39]. The dendrochronological statistics (correlation between series, mean sensitivity; see Table 1) confirm the coherent responses to climate and are similar to those reported in other studies [7,8,11–14]. Nevertheless, the mean sensitivity of *P. lentiscus* (0.39) was much higher than previously reported [13] which is explained because we studied a site under semi-arid conditions with high interannual precipitation variability. We also observed significant positive responses of growth to high winter precipitation of the species inhabiting semi-arid sites (*P. lentiscus*, *C. australis*) and also in *A. unedo* (Figures 3 and 4 and Table 4). This signal may be related to a recharge of soil moisture prior to the spring growth resumption [29] or to the storage of carbohydrates, albeit winter photosynthesis rates are usually lower than in summer [40]. The explanation related to photosynthesis rates cannot be used for deciduous species as *C. australis* which sheds leaves acting as a drought-avoider species. This winter precipitation signal was also observed in the other deciduous species, *P. terebinthus*, but in that case it was not significant whilst the influence of spring-summer precipitation on growth was much stronger (Table 4 and Figures 6 and 7a). Such strong reaction of *P. terebinthus* growth to June precipitation agrees with the correlations observed with SPEI (Figure 4), and suggests a peak in latewood production during that month after earlywood and new leaves are formed. This idea should be further tested using phenology and xylogenesis data or analyzing a long series of wood anatomy. In the case of *L. nobilis*, previous-October wet conditions also improved growth which may alleviate cumulative summer

drought stress (Figure 3). This species may regulate hydraulic conductivity and maintain leaf water supply and high gas exchange rates through xylem embolism repair by using starch stored in the xylem parenchyma [41]. Such resilience could explain its low growth responsiveness to drought (Figure 7e; Table 4), as compared with the other species, despite its growth depending on adequate June precipitation. In the case of *P. lentiscus*, its tight stomatal regulation allows the withstanding of drought stress [13], and the constraining of its cambial dynamics by low water supply in summer and by low temperatures in winter [42] explain its growth sensitivity to drought (Figure 3; Figure 4; Figures 5 and 7b; Table 3; Table 4). This species would follow a drought-avoidance, isohydric strategy by water spending when precipitation increases and allows rapid use of water stored in the soil and radial growth.

The strong growth responses to drought found during June in the two *Pistacia* species studied (Figure 3) agree with a maximum cambial activity in spring [23,42], even if they display contrasting leaf phenology and wood type. The earlier response to drought observed in *C. australis* (May; cf. Figure 4; Figure 7d) may be explained by its deciduousness, making this species dependent on sufficient previous-winter precipitation to form the earlywood and the new leaves. Contrastingly, the growth response of the evergreen *A. unedo* was observed later (July; cf. Figure 4; Figure 7c) suggesting a longer growing season, including growth in autumn, as indicated by some studies on its cambial phenology [42]. Nonetheless, growth in this species was also very dependent on spring precipitation supporting the links between soil water availability, primary growth (leaf and shoot formation), and cambial activity. Lastly, in *L. nobilis* the growth responses to the drought SPEI index were lower than in the other species and rather stable between spring and early summer (Figure 4; Figure 7e). As mentioned before, this species shows a drought-tolerant strategy which allows adequate leaf water status and photosynthesis rates despite a pronounced loss in hydraulic conductivity [41]. This evergreen species is restricted to sites with mild, wet conditions and a short drought, resembling subtropical Tertiary "laurel" forests [18]. These are usually sites located near the coast where low temperatures and frosts are rare and do not reduce photosynthesis activity [43]. This could explain the positive growth response observed in response to high previous-December temperature in *L. nobilis*, which is similar to the positive responses to previous-November temperature observed in *P. lentiscus* (Figure 3). However, no significant positive correlation was found between growth and previous-winter temperatures in the third evergreen species, *A. unedo*, but an almost significant correlation in February was found (Figure 3).

The "*climwin*" analyses allowed the refining of the climate–growth correlation analyses (Figure 5; Figure 6). First, the fitted linear models evidenced the much higher growth variance explained by precipitation (on average 37%) as compared with temperature (13%). Second, the selected climate windows included the spring-to-summer main growing season in agreement with studies on cambium phenology in some of these species as *P. lentiscus* or *A. unedo* [42]. In both climate variables, temperature and precipitation, the median month of the selected climate windows was May. Third, the use of cross-validation and randomization tests refined these results, with mean growth variances for precipitation and temperature of 30% and 7%, respectively, and June as a median month. These refined models only provided significant support to the precipitation model in the case of *P. terebinthus* (Table 3, Figure 6b), confirming that the growth of this species is a robust proxy of hydroclimate and changes in water availability in agreement with the SPEI-growth correlations (Figure 4; Figure 7a). The randomization methods did not provide support to the precipitation–growth model in the case of *P. lentiscus* (Table 3, see the histogram of ΔAICc values and the ΔAICc of the best model fitted in Figure 5b). In both *Pistacia* species, the scatters between the climate (precipitation) and response (ring-width index) variables were in agreement with the assumption of linear relationships (Figure 5; Figure 6). In addition, the *P. terebinthus* chronology was the only one showing a significant correlation with the *L. nobilis* chronology (Table 2) indicating similar growth responses to large-scale climatic patterns over the Iberian Peninsula. The *A. unedo* precipitation model was almost significant, which is in line with the high correlations found in this species with April precipitation and with the July

6-month SPEI. This finding agrees with previous studies showing the sensitivity of this species to spring and summer precipitation [14–16].

The drought–growth "*climwin*" analyses selected the 9-month period as the most important (lowest ΔAICc values, highest $R^2$ values) in the case of all species, except in *A. unedo* and *L. nobilis* (Table 4 and Figure 7). This was in agreement with the correlation analyses (Figure 4). In *A. unedo*, the 1- and 6-month periods were the most important (Figure 4; Figure 7c, Table 4) indicating that his species is more sensitive to short droughts than the other species and confirming its dependence on early-season water availability. Lastly, *L. nobilis* presented a low sensitivity to drought and no "*climwin*" model was significant (Figure 4; Figure 7e, Table 4), which agrees with the mesic conditions of its habitat. Overall, correlation and "*climwin*" analyses based on the SPEI confirmed that *P. lentiscus* and *P. terebinthus* were the most responsive species to changes in water availability. Furthermore, "*climwin*" analyses selected the growing-season period (May and June) in the models with highest predictive power *(P. lentiscus, P. terebinthus)*, but selected longer periods in species showing a more delayed growth response to drought such as *A. unedo* (Figure 4; Figure 7, Table 4).

The "*climwin*" analyses also reported some results, with little biological support, as the selection of October in the climate window of the *L. nobilis* temperature model (Table 3). Nonetheless, this model had a very low explanatory power and it was not significant. Longer and more replicated tree-ring-width series or complementary data (e.g., leaf phenology data, dendrometer records or xylogenesis series, wood anatomy, or isotope data) could be used in future studies to evaluate the utility of similar information-theoretic approaches to filter climate–growth relationships as those used by the "*climwin*" package.

## 5. Conclusions

To conclude, we found that wet-cool conditions during the previous winter and during the current spring enhanced growth in five broadleaf Mediterranean tree and shrub species (*P. terebinthus*, *P. lentiscus*, *A. unedo*, *C. australis,* and *L. nobilis*) which evolved under pre-Mediterranean climate conditions characterized by a weaker seasonality and a less marked summer drought. Growth was constrained by mid to long droughts (9- and 6-month SPEI) during the growing season (May to July) with notable differences between species showing a higher (*P. terebinthus*, *P. lentiscus*) or lower responsiveness (*L. nobilis*) to water shortage. The analyses based on "*climwin*" provided a more conservative characterization of climate–growth relationships showing that only the precipitation model in *P. terebinthus* and considering the climate window from April to July was significant. Some of the possible effects of continued climatic drying on these species would include a reduced radial growth and the contraction of populations inhabiting xeric sites. Finally, we encourage other researchers to compare climate–growth correlation analyses, widely used in dendroecology and other tree-ring disciplines, with information-theoretic approaches such as those used by "*climwin*" and similar statistical packages.

**Supplementary Materials:** The following are available online at http://www.mdpi.com/1999-4907/11/12/1250/s1, Table S1: Summary of the main statistics of selected climate-growth models based on "*climwin*" analyses. Significance levels: * $p < 0.05$; ** $p < 0.01$; *** $p < 0.001$. Table S2: Summary of the main statistics of selected SPEI-growth models based on "*climwin*" analyses. The coefficients of the two SPEI time scales whose models showed the lowest ΔAICc for each species are presented. Significance levels: * $p < 0.05$; ** $p < 0.01$; *** $p < 0.001$.

**Author Contributions:** Conceptualization, J.J.C. and Á.R.-C.; methodology, Á.R.C. and J.J.C.; software, Á.R.-C. and J.J.C.; validation, J.J.C. and Á.R.-C.; formal analysis, Á.R.-C. and J.J.C.; data curation, J.J.C.; writing—original draft preparation, J.J.C. and Á.R.-C.; writing—review and editing, all authors; funding acquisition, J.J.C. All authors have read and agreed to the published version of the manuscript.

**Funding:** This research was funded by Spanish Ministry of Science, Innovation, and Universities, grant number FORMAL (RTI2018-096884-B-C31).

**Acknowledgments:** We thank several colleagues for their assistance during field ad laboratory work. We acknowledge the reviewers and the editor for improving a previous version of the manuscript.

**Conflicts of Interest:** The authors declare no conflict of interest. The funders had no role in the design of the study; in the collection, analyses, or interpretation of data; in the writing of the manuscript, or in the decision to publish the results.

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
