# Peer review of "Relating Climate, Drought and Radial Growth in Broadleaf Mediterranean Tree and Shrub Species: A New Approach to Quantify Climate-Growth Relationships"

_forests, doi:10.3390/f11121250_

Round 1

Reviewer 1 Report

Review of:

Relating climate, drought and radial growth in broadleaf Mediterranean tree and shrub species: new approaches to quantify climate-growth relationships

Julio Camarero, and Álvaro Rubio-Cuadrado

For Journal: Forests

General Comments:

This manuscript presents a concise and valuable study on (1) the climate-growth responses of five generally understudied (dendrochronology) pre-Med. species, and (2) presenting the use of an existing R package climwin in dendrochronological studies. The study is well designed, with clear context, hypotheses, objectives, results and interpretations presented. Generally, the manuscript is well-written, however the Introduction contains may grammatical errors and poor English phrasing at times. Language editing of the Introduction is required. I have included some corrections (see below) but I couldn’t catch them all.

I strongly recommend the inclusion of a map in the body of the manuscript. It would be nice to see a 5 panel map, one panel per species, with the species distribution marked on map (this is not provided in text). Or a single map with all sites and a climate variable interpolated in background (e.g., output from climateexplorer or similar). I also suggest the inclusion of a figure (perhaps in the supplements) that includes several pictures of the tree species. These species may be ones that the dendro community is not terribly familiar with and would benefit from some images to go along with the study.

There are no references to the figures in the Discussion. This makes evaluating the interpretations cumbersome and difficult. I recommend reworking the long text included in the Methods section that describes the climwin output into the Discussion, with interpretations integrated with direct reference to the figures. I also suggest considering some sort of visual that compares the outputs between the ‘regular’ Pearson correlation approach and the climwin approach. Although this may not be practical, given the differing nature of the inputs and outputs, something could be tabulated so the improvements made by climwin can be more easily gleaned.

I enjoyed reading this paper very much and I learned a lot – especially from the Discussion of the possible physiological mechanisms that may underpin the varying drought-growth responses among the species.

Specific Comments:

Abstract:

Line 14: change ‘as’ to ‘including’

Line 15/16: Sentence structure awkward. Change ‘despite’ to ‘, however’

Line 19: why is drought separated from climate?

Line 29: change ‘excepting’ to ‘except’ (line 95 too!)

Line 34/35: Sentence beginning ‘We introduced the use of…’ isn’t necessary, suggest removing. Instead, work it into the next sentence (e.g., We argue that climwin and similar information….’)

Keywords: I am unsure of how many key words you can use, but I would suggest including dendrochronology, dendroclimatology, Mediterranean, tree rings

Introduction:

Line 44: change ‘or’ to ‘and’

Line 47: how much have temps increased, quantify statement and give temporal interval (e.g., temperatures have rise X degrees Celsius over the past XX years)

Line 49: remove ‘particularly’

Lines 46-49: content repetitive, editing needed

Line 49: ‘a better knowledge’ doesn’t make sense – we need ‘better knowledge’

Line 51: allows a better understanding

Line 53: ‘usually width but could be others…’ change this to ‘usually ring width but also density, ….’

Line 57: change ‘showing’ to ‘with’, change ‘which’ to ‘that’

Line 64: incorrect use of ‘as’

Line 69: ‘evolutionary original’ – do you mean ‘evolutionary origins’?? – also remove ‘historical’ as these constraints occur today too right?

Line 77: change ‘that’ to ‘this’

Line 78: I think a bit more is needed here to situated the general use of this package since 2016 – and why you think it could be well-applied to tree-ring datasets.

Materials and Methods:

Line 91: move ‘growth’ to after ‘year-to-year’

Line 139: Does this methodology apply to all sites? If so, maybe start section with ‘At all sites..’

Line 144: check journal standards, but generally numbers less than ten are written out

Line 151: remove ‘by’

Line 163: considered is spelt wrong

Line 169: ‘We decided presenting correlations…’ is poor writing. ‘We present correlations with precipitation…’ is better.

Line 176: How were significance thresholds calculated?

Line 180: ‘we tested an annual time’ – what does this mean?

Line 178-209: This is a lot of text dedicated to describing the output of an existing R package. Could this text not be great reduced with citations to original presentation of the package and relevant applications as examples?

Results:

Table 1: could a subscript be added to the table heading ‘best replicated period’ to briefly explain how this was determined

Line 237: ‘which’ should be used with a ‘,’ – otherwise use ‘that’

Table 2: put stars next to significant correlation values at the 0.05 level so I don’t have to look back and forth between correlation value and sig level.

Figure 1. I am not sure about the importance of this figure. I would move to supplements, unless you plot some climate data on it too to add another dimension. Also, what’s plotted here doesn’t indicate the best replicated period outlined in Table 1. Maybe add some marker to highlight the best replicated period so it can be compared to sample depth? Otherwise I would move this figure to the supplements and add the temporal span of chronologies to Table 1.

Table 3. Add horizontal lines separating species.

Discussion:

Figures 4 and 5. Very LITTLE direct interpretation of these panel figures is provided. I suggest working in references to the figures and better explanation of the output and how it relates to findings/interpretations. Perhaps rework the text in the methods that describes the climwin output into the discussion and integrate the interpretation of the results. This would be much more useful to the reader. Rather than having to go back to the methods to figure out what each graph means. At the very least, direct reference to the Figures and specific panels is needed so I can connect the interpretations in the Discussion to what is presented in Figures 4 and 5.

Line 319: incorrect use of ‘as’

Line 339: do you mean ‘resilience’ instead of ‘resilient’?

Generally excellent interpretation of physiological reasoning behind climate-growth responses. I learned a lot from this section.

Conclusion:

Consider adding a sentence about the possible effects of continued climatic drying on these species in the future?

Author Response

Comments and Suggestions for Authors

Reviewer 1: Relating climate, drought and radial growth in broadleaf Mediterranean tree and shrub species: new approaches to quantify climate-growth relationships

Julio Camarero, and Álvaro Rubio-Cuadrado

General Comments:

This manuscript presents a concise and valuable study on (1) the climate-growth responses of five generally understudied (dendrochronology) pre-Med. species, and (2) presenting the use of an existing R package climwin in dendrochronological studies. The study is well designed, with clear context, hypotheses, objectives, results and interpretations presented. Generally, the manuscript is well-written, however the Introduction contains many grammatical errors and poor English phrasing at times. Language editing of the Introduction is required. I have included some corrections (see below) but I couldn’t catch them all.

We have revised the Introduction following your comments and improved the English expressions.

I strongly recommend the inclusion of a map in the body of the manuscript. It would be nice to see a 5 panel map, one panel per species, with the species distribution marked on map (this is not provided in text). Or a single map with all sites and a climate variable interpolated in background (e.g., output from climate explorer or similar). I also suggest the inclusion of a figure (perhaps in the supplements) that includes several pictures of the tree species. These species may be ones that the dendro community is not terribly familiar with and would benefit from some images to go along with the study.

Following your comments we have included a map of the studied locations and some images of the sampled species. Maps were built using GBIF occurrences (https://www.gbif.org) since some maps are available on the web for some species but not for the five studied species. We have also included images of the sampled sites or species.

There are no references to the figures in the Discussion. This makes evaluating the interpretations cumbersome and difficult. I recommend reworking the long text included in the Methods section that describes the climwin output into the Discussion, with interpretations integrated with direct reference to the figures. I also suggest considering some sort of visual that compares the outputs between the ‘regular’ Pearson correlation approach and the climwin approach. Although this may not be practical, given the differing nature of the inputs and outputs, something could be tabulated so the improvements made by climwin can be more easily gleaned.

We added references to the figures and tables in the revised Discussion. We summarized the climwin analyses in Tables to allow readers comparing the Pearson correlations with the climwin outputs and find out the climwin improvements.

I enjoyed reading this paper very much and I learned a lot – especially from the Discussion of the possible physiological mechanisms that may underpin the varying drought-growth responses among the species.

Thank you.

Specific Comments:

Abstract:

Line 14: change ‘as’ to ‘including’

Line 15/16: Sentence structure awkward. Change ‘despite’ to ‘, however’

Line 19: why is drought separated from climate?

Line 29: change ‘excepting’ to ‘except’ (line 95 too!)

Line 34/35: Sentence beginning ‘We introduced the use of…’ isn’t necessary, suggest removing. Instead, work it into the next sentence (e.g., We argue that climwin and similar information….’)

Keywords: I am unsure of how many key words you can use, but I would suggest including dendrochronology, dendroclimatology, Mediterranean, tree rings

We performed all suggested changes.

Introduction:

Line 44: change ‘or’ to ‘and’

Line 47: how much have temps increased, quantify statement and give temporal interval (e.g., temperatures have rise X degrees Celsius over the past XX years)

Line 49: remove ‘particularly’

Lines 46-49: content repetitive, editing needed

Line 49: ‘a better knowledge’ doesn’t make sense – we need ‘better knowledge’

Line 51: allows a better understanding

Line 53: ‘usually width but could be others…’ change this to ‘usually ring width but also density, ….’

Line 57: change ‘showing’ to ‘with’, change ‘which’ to ‘that’

Line 64: incorrect use of ‘as’

Line 69: ‘evolutionary original’ – do you mean ‘evolutionary origins’?? – also remove ‘historical’ as these constraints occur today too right?

Line 77: change ‘that’ to ‘this’

Line 78: I think a bit more is needed here to situated the general use of this package since 2016 – and why you think it could be well-applied to tree-ring datasets.

We edited and corrected all commented sentences and words. We added some explanation to justify the use of the “climwin” packaged in tree-ring sciences. Specifically, we emphasized that this package allows objectively defining the relevant climate window for tree growth, which is very important for tree-ring sciences.

Materials and Methods:

Line 91: move ‘growth’ to after ‘year-to-year’

Line 139: Does this methodology apply to all sites? If so, maybe start section with ‘At all sites..’

Line 144: check journal standards, but generally numbers less than ten are written out

Line 151: remove ‘by’

Line 163: considered is spelt wrong

Line 169: ‘We decided presenting correlations…’ is poor writing. ‘We present correlations with precipitation…’ is better.

Line 176: How were significance thresholds calculated?

Line 180: ‘we tested an annual time’ – what does this mean?

Line 178-209: This is a lot of text dedicated to describing the output of an existing R package. Could this text not be great reduced with citations to original presentation of the package and relevant applications as examples?

We edited and corrected all commented sentences and words. Significance thresholds were calculated by comparing with reference distributions. Following your suggestion, we also reduced the text describing the climwin package by referring to original citations and presentation of the package.

Results:

Table 1: could a subscript be added to the table heading ‘best replicated period’ to briefly explain how this was determined

Line 237: ‘which’ should be used with a ‘,’ – otherwise use ‘that’

Table 2: put stars next to significant correlation values at the 0.05 level so I don’t have to look back and forth between correlation value and sig level.

Figure 1. I am not sure about the importance of this figure. I would move to supplements, unless you plot some climate data on it too to add another dimension. Also, what’s plotted here doesn’t indicate the best replicated period outlined in Table 1. Maybe add some marker to highlight the best replicated period so it can be compared to sample depth? Otherwise I would move this figure to the supplements and add the temporal span of chronologies to Table 1.

Table 3. Add horizontal lines separating species.

We edited and corrected all commented sentences and words. We also edited the Tables as suggested. Regarding Figure 1 we prefer to keep it to show the year-to-year growth variability of the study species and this was commented by reviewer 2, who indicated strengthening the description and presentation of tree-ring methods, including cross-dating of samples. We have modified it indicating the best replicated period.

Discussion:

Figures 4 and 5. Very LITTLE direct interpretation of these panel figures is provided. I suggest working in references to the figures and better explanation of the output and how it relates to findings/interpretations. Perhaps rework the text in the methods that describes the climwin output into the discussion and integrate the interpretation of the results. This would be much more useful to the reader. Rather than having to go back to the methods to figure out what each graph means. At the very least, direct reference to the Figures and specific panels is needed so I can connect the interpretations in the Discussion to what is presented in Figures 4 and 5.

We agree on your remark. We have added more interpretation of these figures. We have better integrated the figures interpretation by using the text in methods explaining the “climwin” output. We have also added direct references to the Figures and panels in the revised Discussion as you suggested. We have followed a similar approach regarding the new drought-growth “climwin” analyses provided in Table 4 and Figure 7.

We consider that these changes have improved and strengthened the Discussion by providing a comprehensive comparison of correlations and “climwin” analyses.

Line 319: incorrect use of ‘as’

Line 339: do you mean ‘resilience’ instead of ‘resilient’?

We revised and corrected both sentences.

Generally excellent interpretation of physiological reasoning behind climate-growth responses. I learned a lot from this section.

Thank you.

Conclusion:

Consider adding a sentence about the possible effects of continued climatic drying on these species in the future?

Thank you for the suggestion. We did it.

Thanks a lot for your constructive review.

Reviewer 2 Report

The manuscript (MS) describes dendroclimatic analyses of five tree-ring chronologies of different broadleaf species using conventional and novel (for dendroclimatic studies) methodology. It is well structured and written (just needs minor language corrections). It is brief, but at the same time contain necessary information. I liked the introduction, and the methods sections a lot. The study has a well-posed hypothesis that is tested using an application of “climwin” analyses. The strong side of the MS is a good description of the sampled trees and sites, as well as species origins and peculiar properties of their physiology.

I think that the introduction of “climwin” for dendroclimatological analysis is a valuable contribution. Indeed, the authors showed that it can add important information on climatic controls of tree growth, as compared to the traditional methods.

However, I have several major concerns about the MS.

First, the authors did not mention the use of COFECHA program in their methodology. This program is widely used for the control of measurements quality and cross-dating in dendrochronology. Though some of the results suggest that the chronologies are well-dated (e.g. good correlations of the chronologies of different species), still some of the tree-ring series or parts of the series might be misdated, which may reduce the quality of the final chronologies.

Second, “climwin” was initially designed for high-resolution (e.g. daily) climate series. In the MS authors only show results of the “climwin” analysis using monthly variables (temperature and precipitation), but do not include results of the “climwin” analysis using 7-day SPEI data. These results would be important first to estimate climatic window of the primary factor affecting tree growth in the region, and second to further understand the utility of “climwin” for dendroclimatological analysis.

Third, it is not clear which linear models are finally used in “climwin”. How are the climatic predictors are used? Are they averaged or used separately with different coefficients? What are the coefficients? Is it possible to get this information from the program? It would be good to see the descriptions of the selected models at least as a supplementary.

Forth, I think more discussion is needed on the results of “climwin”. The significance of the models with and without cross-validation is quite different in some cases. Is it due to overfitting or you have another explanation? Might the results that are not significant on cross-validation be used or not? Also, a detailed discussion of every of the six panels of “climwin” graphical output (Fig 4 and 5) will be good for the newly introduced methodology.

Several minor points:

Section 2.1. The first sentence is not clear and badly written. Climatically sensitive sites or trees? How did you know it before the dendroclimatic analysis? Was it expected somehow?  

Section 2.1. The last paragraph: “excepting in the L. nobilis site” – what is happening in this site?

Page 4, line 4: “this species may for wedging rings” – a word is missing?

Page 4, line 6: “5.15 mm-wide” – I think you mean core diameter.

Section 2.3, the last paragraph: “by subtracting observed from fitted values”. Usually variance stabilization is used when difference is selected over the more common ratios. Did you use variance stabilization?

Section 2.3.The last sentence: “We considred used an expressed” – correct the wording

Section 2.4, the first sentence: “related to” – “is” is missing?

Section 2.5. What are the “beta-estimates of the effect of the climate signal on growth”? Please explain it in more details.

Section 3.3, line 5: “followed by to precipitation in the case of A. unedo” – something is wrong here

To summarize, if the COFECHA program was not used (not just forgotten in the description), I recommend checking all the measurements and the results of cross-dating with it. Otherwise the results might be flawed.

Also, I recommend a major revision with inclusion of additional results on drought-growth interactions using “climwin”, and additional discussion of the obtained “climwin” results.

Author Response

Reviewer 2: Relating climate, drought and radial growth in broadleaf Mediterranean tree and shrub species: new approaches to quantify climate-growth relationships

Julio Camarero, and Álvaro Rubio-Cuadrado

Comments and Suggestions for Authors

The manuscript (MS) describes dendroclimatic analyses of five tree-ring chronologies of different broadleaf species using conventional and novel (for dendroclimatic studies) methodology. It is well structured and written (just needs minor language corrections). It is brief, but at the same time contain necessary information. I liked the introduction, and the methods sections a lot. The study has a well-posed hypothesis that is tested using an application of “climwin” analyses. The strong side of the MS is a good description of the sampled trees and sites, as well as species origins and peculiar properties of their physiology.

I think that the introduction of “climwin” for dendroclimatological analysis is a valuable contribution. Indeed, the authors showed that it can add important information on climatic controls of tree growth, as compared to the traditional methods.

However, I have several major concerns about the MS.

First, the authors did not mention the use of COFECHA program in their methodology. This program is widely used for the control of measurements quality and cross-dating in dendrochronology. Though some of the results suggest that the chronologies are well-dated (e.g. good correlations of the chronologies of different species), still some of the tree-ring series or parts of the series might be misdated, which may reduce the quality of the final chronologies.

We always use COFECHA to check our visual cross-dating. The program is cited in revised version of the ms. We also calculated several dendrochronological statistics (e.g., EPS) showing that our chronologies are well replicated and the individual series are reliably cross-dated.

Second, “climwin” was initially designed for high-resolution (e.g. daily) climate series. In the MS authors only show results of the “climwin” analysis using monthly variables (temperature and precipitation), but do not include results of the “climwin” analysis using 7-day SPEI data. These results would be important first to estimate climatic window of the primary factor affecting tree growth in the region, and second to further understand the utility of “climwin” for dendroclimatological analysis.

As suggested, we added the analyses related to SPEI data in a new section and discussed them.

Third, it is not clear which linear models are finally used in “climwin”. How are the climatic predictors are used? Are they averaged or used separately with different coefficients? What are the coefficients? Is it possible to get this information from the program? It would be good to see the descriptions of the selected models at least as a supplementary.

We used the linear models plotted in Figures 5 and 6, and also in Figure 7 in the case of the SPEI. Climate predictors are averaged. We have provided the description of the selected models in the supplementary information (Tables S1 and S2).

Forth, I think more discussion is needed on the results of “climwin”. The significance of the models with and without cross-validation is quite different in some cases. Is it due to overfitting or you have another explanation? Might the results that are not significant on cross-validation be used or not? Also, a detailed discussion of every of the six panels of “climwin” graphical output (Fig 4 and 5) will be good for the newly introduced methodology.

As you and the reviewer 1 suggested, we provided a more in-depth description and discussion of the “climwin” analyses presented in Figures 4 and 5.

Several minor points:

Section 2.1. The first sentence is not clear and badly written. Climatically sensitive sites or trees? How did you know it before the dendroclimatic analysis? Was it expected somehow?  

We rephrased the sentence. We selected sites with conditions to strengthen climate-growth associations such as xeric locations.

Section 2.1. The last paragraph: “excepting in the L. nobilis site” – what is happening in this site?

We rephrased the sentence. In this site winter precipitation is higher than spring precipitation.

Page 4, line 4: “this species may for wedging rings” – a word is missing?

We rephrased the sentence: “this species may form wedging rings”.

Page 4, line 6: “5.15 mm-wide” – I think you mean core diameter.

We rephrased the sentence. You are right.

Section 2.3, the last paragraph: “by subtracting observed from fitted values”. Usually variance stabilization is used when difference is selected over the more common ratios. Did you use variance stabilization?

Sorry, this is a mistake. We calculated indices by using ratios and did not use variance stabilization. We rewrote this part.

Section 2.3.The last sentence: “We considred used an expressed” – correct the wording

We corrected it.

Section 2.4, the first sentence: “related to” – “is” is missing?

We corrected it and added “is”. Thank you.

Section 2.5. What are the “beta-estimates of the effect of the climate signal on growth”? Please explain it in more details.

We explained it. The beta estimates are the coefficient estimates (slopes of the linear models) for the climate variable terms in the models of tree growth variability.

Section 3.3, line 5: “followed by to precipitation in the case of A. unedo” – something is wrong here

We corrected it by adding the months when the precipitation signal was strongest.

To summarize, if the COFECHA program was not used (not just forgotten in the description), I recommend checking all the measurements and the results of cross-dating with it. Otherwise the results might be flawed.

We always use COFECHA to check our visual cross-dating. The program is cited in revised version of the ms. Thanks for reminding us to cite it.

Also, I recommend a major revision with inclusion of additional results on drought-growth interactions using “climwin”, and additional discussion of the obtained “climwin” results.

We added the drought-growth interactions and further discussed the “climwin” analyses. Thanks a lot for your constructive review.

Round 2

Reviewer 2 Report

The authors addressed almost all of my comments, and added the information that I had considered to be lacking. The added results about response to SPEI, to my mind, improved the MS.